# Developing a Measuring System for Monitoring the Thickness of the 6 m Wide HDPE/LDPE Polymer Geomembrane with Its Continuous Flow Using Automation Equipment

**Tatyana Nikonova [1,*]**, **Olga Zharkevich [1]**, **Essim Dandybaev [1]**, **Murat Baimuldin [2]**, **Leonid Daich [3]**, **Andrey Sichkarenko [3]** and **Evgeniy Kotov [3]**

[1] Department of Technological Equipment, Mechanical Engineering and Standardization, Karaganda Technical University, Karaganda 100027, Kazakhstan; zharkevich82@mail.ru (O.Z.); esim.dandybayev@list.ru (E.D.)

[2] Department of Mineral Deposit's Development, Karaganda Technical University, Karaganda 100027, Kazakhstan; murat_owl@mail.ru

[3] Department of Automated Manufacturing Processes, Karaganda Technical University, Karaganda 100027, Kazakhstan; l.daych@mail.ru (L.D.); sichkarenko@gmail.com (A.S.); kotov1988@mail.ru (E.K.)

**\*** Correspondence: nitka82@list.ru

**Abstract:** As a result of R&D, a measuring system for controlling the thickness of the HDPE/LDPE (high-density polyethylene/low-density polyethylene) polymer geomembrane was developed using automation equipment. The relevance of this work consists of the development of a domestic, relatively inexpensive system for controlling the thickness of the HDPE/LDPE polymer geomembrane in production conditions based on modern equipment for enterprise automation. The scientific novelty consists of the use of original design solutions in the development of hardware and software complex mechanisms that allow controlling the thickness of the HDPE/LDPE polymer geomembrane layers along the entire width of the shaft, excluding deformation of the film as a result of foreign bodies entering during extrusion, cuts, monitoring the quality of the film in real time, as well as the possibility of analyzing the measured parameters in the database of the automated system.

**Keywords:** metrology systems; measurement; laser sensor; automatic control; polymer geomembrane; high-density polyethylene; low-density polyethylene

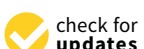



## 1. Introduction

One of the main products made of polymeric materials is polyethylene geomembrane [1]. The quality of the polymer geomembrane is affected by several probable reasons, which are associated with the original materials, extrusion technology, and equipment operation [2,3]. In conditions when the quality of the property of polyethylene products suddenly decreases, the presence of unusable products in their manufacturing increases, the external characteristics of the geomembrane deteriorate, and the strength of the material also decreases with unchanged basic parameters of equipment and temperature [4,5]. All these factors lead to a significant decrease in productivity and unnecessary costs. As a result of the timely elimination of local causes, the subsequent stability of production is unpredictable, and after an indefinite period time, the situation is possible in which deterioration of the film quality begins again and productivity decreases [6,7].

There are a lot of factors that affect fluctuations in the properties of HDPE/LDPE polymer geomembranes in the production environment. This is caused by changes in temperature and gas conditions, imperfection of technological processes of extrusion, removal, and cooling of the manufactured material [8,9].

For customers of polymer films, the following indicators are important [10,11]:

(1) quality;
(2) strength;

(3)    rigidity;
(4)    the ability to resist puncture and tearing;
(5)    flatness;
(6)    uniformity.

The production and quality of the HDPE/LDPE polymer geomembrane is controlled by GOST R 56586-2015 [12]. This standard applies to extruded polyethylene geomembrane made of high-density polyethylene. The film is used in agriculture, in reclamation environments, and in construction; it is also used as a material for packaging in various sectors of the national economy [13].

The geomembrane is produced in rolls in the form of a sleeve, a half-sleeve (a sleeve cut along the entire length on one side), a sheet (a sleeve cut along the entire length on both sides with or without trimming edges), folded sleeves (with folds), half-folded sleeves, and other types.

One of the main qualities of the HDPE/LDPE polymer geomembrane for the production of agricultural hoses is its strength [13]. It is achieved through the use of several different layers, as well as ensuring the specified parameters of the layer thickness, which is characteristic of today's composite materials [14]. Controlling the thickness of the geomembrane layers, as well as the rejects in the case of the ingress of various foreign bodies during extrusion, is carried out visually or using thickness gauges of various designs and physical principles [15,16]. Industrial automated thickness measurement systems are not produced in Kazakhstan. However, there is a growing trend in the world of using optical measuring systems supporting implementation of industrial processes in machines where such systems have not been used before [17,18].

There is a system of measuring the thickness of a moving geomembrane (measuring system of the LAP-Laser company, CALIX) containing a supporting structure made in the form of a frame with a supporting surface, two laser triangulation sensors rigidly fixed to the frame, an electronic unit, and a power supply [19]. The thickness of the geomembrane is calculated based on the values of the distances from the geomembrane to each of the sensors. Measurement data can be displayed and processed using an electronic unit, a programmable controller, or a personal computer. The disadvantage of this system (the CALIX system) is the presence of two laser sensors, which complicates the system for measuring the thickness of a moving geomembrane and lowers the accuracy of measuring the thickness of the geomembrane.

The measuring system proposed by Yu.A. Sazonov and S.V. Zhigalkin is also widely used [20]. The system contains a supporting structure made in the form of the frame with a sliding surface and one laser triangulation sensor installed on it, an electronic data processing unit and a power supply unit, characterized in that the frame and the laser sensor are rigidly fixed to it form the measuring unit with a constant distance from the sensor to the sliding surface. The disadvantage of the system is that measurements of the distance to the friction surface are carried out only at one point, while the microrelief of the friction surface is not taken into account, and the measurements of the geomembrane thickness are made in a small section of the geomembrane in relation to its width, which leads to additional errors in measuring the geomembrane thickness.

There are also such measuring systems as geomembrane thickness gauge of the KAPA series (Austria) [21], the SHADOW series thickness gauge (Austria) [22], the Platforma 21 systems! and Platforma IPlus! (Russia) Thermo Scientific [23], the Controlplast system (Russia) [24], the Sintel 9000 system (Italy) [25], and the OVEN geomembrane thickness control system (Russia) [26]. The characteristics of the above systems are presented in Table 1.

**Table 1.** Characteristics of measuring systems.

| Industrial Thickness Gauge Name | Metrological Characteristics |
|---|---|
| Thickness gauge for geomembranes, KAPA series (Austria) | Measuring width: 400–3000 mm<br>Thickness: up to 2.0 mm<br>Accuracy: 0.1 μm |
| Thickness gages SHADOW series (Austria) | Measuring width: 400–3000 mm<br>Thickness: up to 3.5 mm (5.5mm)<br>Accuracy: 3.0 μm |
| Platforma 21 systems!<br>Platforma IPlus! (Russia) | Measuring width: 400–3000 mm<br>Thickness: up to 2.5 mm (5.5 mm)<br>Accuracy: 2.0 μm |
| Controlplast system (Russia) | Measuring width: 400–3000 mm<br>Thickness: up to 3.5 mm (5 mm)<br>Accuracy: 1.0 μm |
| Sintel 9000 system (Italy) | Measurement width–1700 mm<br>Thickness: 10–200 μm<br>Accuracy: 0.1 μm |
| Geomembrane thickness control system OVEN (Russia) | Measuring width: 400–3000 mm<br>Thickness: up to 3.5 mm (5 mm)<br>Accuracy: 1.0 μm |

Significant disadvantages of the existing models of industrial control systems are high manufacturing costs or high measurement errors. The cost of such systems reaches tens of thousand dollars. So, for example, the price of the capacitive geomembrane thickness measuring system Kundig's K-100 Twin reaches USD 100,000. If the system is supplemented with an actuator for controlling the geomembrane thickness, then its price can reach EUR 200,000. At the same time, in the USA, at least a hundred such systems are installed per year, according to John Wise, a representative of the German Reifenhauser company in the USA.

Thus, the need to solve the production problem of designing and manufacturing an automated system for controlling the thickness of the geomembrane in the process of its manufacture is urgent.

## 2. Materials and Methods

The Altyn Arna Holding LLP is a Kazakhstan domestic manufacturer of polyethylene products, geomembrane and geosynthetic materials, polyethylene packaging, polyethylene large-format heat shrinkage, and polypropylene and polyethylene liners of wagons for transportation of bulk materials [27].

The task is to design an automatic control measuring system for the 6 m wide HDPE/LDPE polymer geomembrane, which implies the use of 3 extruders at once in the production of the geomembrane (Figure 1).

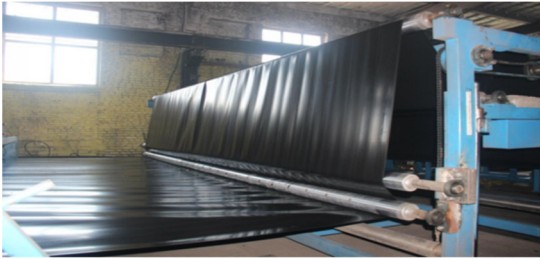

**Figure 1.** 6 m wide polymer geomembrane.

Table 2 presents the input data for the design of a system for measuring the thickness of a web with a width—6 m.

**Table 2.** Parameters for choosing a measurement method.

| Parameter | Value |
|---|---|
| Control type | Nondestructive Continuous In real time |
| Working temperature range, °C | −20 + 30 |
| Time of continuous work, h | 8 |
| Geomembrane transfer speed from the extruder, m/min | 3–5 |
| Display of current thickness | + |
| Accumulation of information, T, period | 1 month |
| Removable media transfer capability | + |
| Measured thickness range, mm | 0.005–5 |
| Accuracy, μm | ±25 |
| Alarm | + |
| Controlled web width, no more, m | 6 |
| Control over the entire width (allowed at several "migrating" points covering the entire width of the product per cycle) | + |

The proposed system decreases the error during continuous measurement of the thickness of a wide geomembrane during its movement, and it obtains information of its thickness violation (Figure 2). The system contains a supporting structure made in the form of a frame with a rotating support roller, an electronic data processing unit, and a power supply unit, as well as a laser sensor mounted on a trolley, which moves parallel to the support roller and perpendicular to the direction of geomembrane movement. A drive is used, and the reversing of the trolley with an electric drive and a laser sensor is carried out according to the signals of the limit switches that fix the normalized geomembrane width.

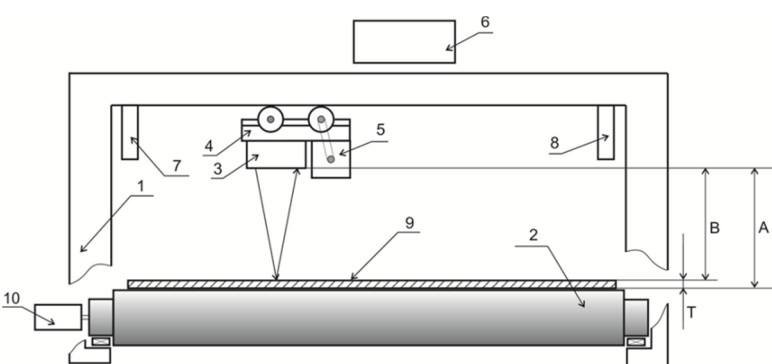

**Figure 2.** Design of an automatic control measuring system for the 6 m wide HDPE/LDPE polymer geomembrane. 1—frame; 2—support roller; 3—laser sensor; 4—carriage; 5—stepper motor; 6—control unit; 7.8—switches; 9—moving geomembrane; 10—rotation angle sensor.

The above set of essential features makes it possible to take into account the microrelief of the friction surface and changes in the thickness of the geomembrane over its entire width, due to scanning with the laser sensor above the support roller and the surface of the geomembrane perpendicular to the direction of geomembrane movement between the limit switches, which increases the accuracy of measuring the thickness of the moving geomembrane over the entire normalized width and obtains information of its thickness violation.

The automatic control system consists of frame 1 with support roller 2 and laser sensor 3 rigidly fixed to carriage 4. Carriage 4 moves along frame 1 parallel to the support roller

2. On carriage 4, drive motor 5 is connected to electronic control unit 6, which is also connected to laser sensor 3. The inputs of the electronic control unit 6 are connected with limit switches 7 and 8, fixed at the ends of the trajectory of movement of carriage 4. Moving geomembrane 9 rests on rotating support roller 2. The angle of rotation of rotating support roller 2 relative to the beam of laser sensor 3 is measured using angle sensor 10, which is connected to electronic control unit 6.

The developed system works as follows. Before starting to measure the thickness of moving polymer geomembrane 9, the entire surface of support roller 2 is pre-scanned by laser sensor 3 by means of moving carriage 4 with laser sensor 3 between limit switches 7 and 8 and rotation of support roller 2. The obtained value of the distance A is stored in electronic unit 6.

The microrelief of the of the rotating support roller 2 violations are fixed in two coordinates. The first coordinate is calculated by the number of control pulses arriving at drive motor 5 from electronic control unit 6. The second coordinate is calculated from the information from rotation angle sensor 10. Regardless of the speed of movement of polymer geomembrane 9, electronic control unit 6 takes into account violations of the microrelief of the friction surface of rotating reference roller 2.

As a result, in the non-volatile memory of control unit 6, the value of the distance from laser sensor 3 to the friction surface of rotating support roller 2 is stored, that is, its microrelief with reference to geometric dimensions and position.

In the operating mode, when geomembrane 9 is introduced into the measurement zone, its tight fit to the friction surface of support roller 2 is achieved, and the value of film 9 thickness is recorded on the indicator of electronic control unit 6.

Scanning is carried out within the normalized width of the geomembrane, since, due to the uneven edges, after controlling its thickness, the edges of the geomembrane are cut off.

The thickness of the geomembrane T is calculated as the difference between the distance A, which is in the memory of electronic control unit 6, and the distance B, measured by laser sensor 3, to the surface of geomembrane 9 in real-time during the movement of geomembrane 9.

The measurement error depends on the parameters of the laser sensor used and the mechanical rigidity of frame 1.

Laser sensor 3 is rigidly attached to carriage 4, which is moving along frame 1; therefore, the distance from sensor 3 to the friction surface of rotating support roller 2 is a known value, taking into account its microrelief.

Electronic control unit 6 provides power and receives the information from limit switches 7 and 8, rotation angle sensor 10, laser sensor 3, and control drive motor 5; it provides a prompt indication of the film thickness value, fixing the change in the geomembrane thickness, the geometric position of the place of violation, as well as real-time detection of thickness violations and sound and light indication of violation detection to attract the attention of maintenance personnel.

The blocks of the device for measuring the thickness of a moving wide polymer geomembrane are made based on known technical solutions as follows.

Frame 1 is made in the form of an aluminum profile AlMgSi, alloy AD-31, with a size of 60 × 60 mm, so that the entire structure has maximum rigidity. Support roller 2 is made in the form of a freely rotating steel roller supported on roller bearings. Laser sensor 3 is made using a commercially available analog-to-digital laser sensor of the LE250IQ type. Movable carriage 4 is made in the form of a platform with 4 wheels and a drive motor. Drive motor 5 for moving the carriage is made using a commercially available 57BYG250C stepper motor.

Electronic control unit 6 is made in the conventional manner based on the known technical solutions and contains a power supply of the J-360-24V type, a stepper motor driver of the TB6600 4A type, and an industrial controller of the OVEN type SPK 110 [M01]. Limit switches 7 and 8 are made using commercially available non-contact inductive

sensors of the KIPPRIBOR LK18M-35.4 type. The angle sensor of support roller 10 is made using a commercially available encoder of the LIR-158B-1-N-2000-10 . . . 20-PI-8 type.

Thus, preliminary measurement and memorization of the microrelief of the rotating support roller without the geomembrane, continuous scanning of the thickness of the moving geomembrane within the normalized width, and calculation of its thickness allows obtaining continuous information of the geomembrane thickness violation and promptly controlling the film extrusion process.

Figure 3 shows a simplified functional diagram of the geomembrane thickness control system.

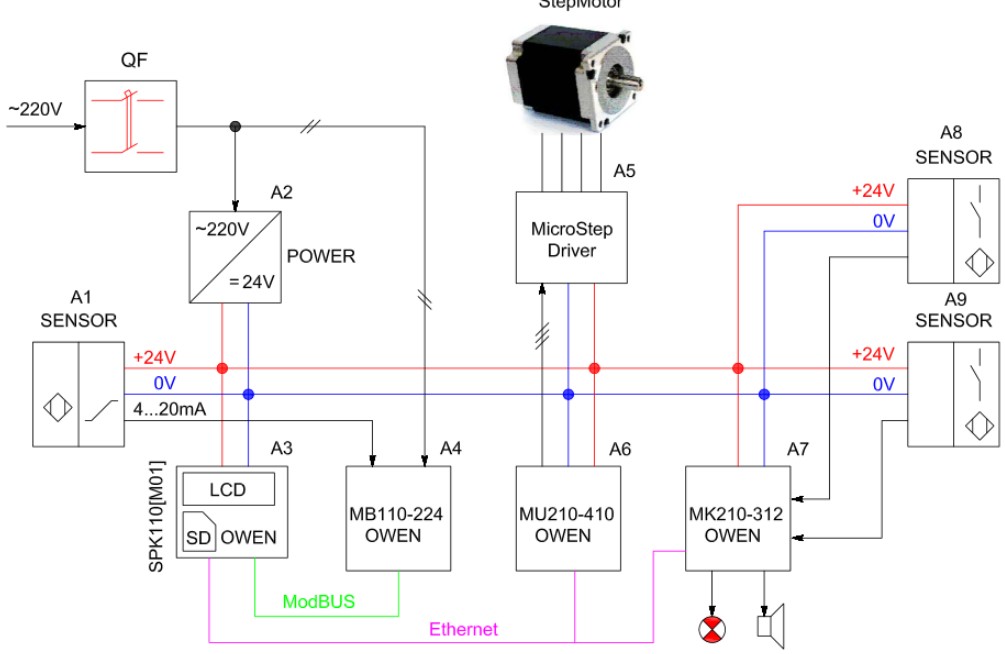

**Figure 3.** Functional diagram of geomembrane thickness control.

As a part of the HDPE/LDPE geomembrane thickness control system, the following main blocks can be distinguished:

A1—an optical distance sensor that monitors the geomembrane thickness;

A2—a DC24V power supply unit supplying power to the controller and system sensors;

A3—an industrial controller with built-in visualization system that controls the entire system;

A4—a module for processing the analog signal from the A1 sensor;

A5—a stepper motor driver moving the measuring carriage with A1 sensor;

A6—a module for controlling the movement of the measuring carriage with the A1 sensor;

A7—a module for input/output of discrete signals from end position sensors and alarm control;

A8, A9—end position sensors of the measuring carriage with the A1 sensor.

All the modules of the system are interconnected by a single information Ethernet, the network for data exchange. In some cases, communication with the modules is carried out via ModBUS.

## 3. Results

The analysis of the results was carried out for the thickness measurement system with one distance sensor and preliminary construction of the base profile.

### 3.1. Building the Base Profile

The base profile is built before each new production cycle of the HDPE/LDPE geomembrane. The periodicity can also be due to external circumstances that affect the change in the geometry of the bending of the base. An example can be the effect of temperature on the bending of the suspension profile of a movable carriage.

When the profile development mode is selected, the carriage moves to any nearest origin of coordinates. The last position of the carriage between the two limit switches that determine the two extreme points of movement is always fixed in the non-volatile memory of the controller.

The procedure for moving the carriage with the laser sensor is started, and the values of the distance from the sensor to the stationary sliding surface are continuously recorded. The cycle of recording the measured value of the distance into the non-volatile memory is 1 s. The discreteness of the movement of the carriage along the width of the geomembrane is 2 cm. With the distance of 6000 mm between the two extreme points of measuring the width of the film, it will take 300 counts and 5 min. The tabular profile data is replaced by the analytically calculated profile change function from the coordinate of the carriage movement.

### 3.2. The Working Cycle of Controlling the Geomembrane Thickness

The working cycle of controlling the geomembrane thickness is carried out from one extreme point determined by the limit switch to the other one located at the opposite end of the membrane.

When the system is started in the "Working cycle" mode, the carriage moves to any nearest origin of coordinates. The discreteness of the displacement of the measuring carriage in the operating mode is 2 mm. Since there is no need to write the data to the FLASH memory in the thickness control mode, measurements can be made every 20 ms. It will take 1 min to monitor 3000 sensor-to-film distance values.

The calculated value of the geomembrane thickness is compared with the value of the permissible deviation from the target value. In case of deviation of the geomembrane thickness beyond the permissible limits, the controller issues a signal to the detector with an audible siren and light indication.

Figure 4 shows the information display screen of the industrial controller, which displays a graph of the current values measured from Frame 1 to the geomembrane. The specified value represents the thickness of the film. In case of deviation of the preset geomembrane thickness, expressed as a distance from Frame 1, a light signal is turned on. The figure shows the range of permissible deviation from the set value of the geomembrane thickness parameters of 60 microns. At the upper boundary of the geomembrane thickness, expressed at a distance of 158,000 microns, there is an increase in thickness by 22 microns. The indicator light will also turn on when values exceed the lower limit of the geomembrane thickness (in Figure 3, the values are above the range of permissible deviation).

Figure 5 shows a graph of the material thickness profile measured with the LE250IQ laser rangefinder.

The analysis of the graph (Figure 6) shows that the measurement result is not a continuous current function of 4–20 mA but discrete readings of current values in the 4–20 mA range (in the example presented, every 1.5 mm).

As a result, "peaks" of discrete values are visible on the graphs of thickness measurement. The measured thickness profile is more likely to be guessed than unambiguously recorded against the background of peaks. The overall picture of the profile change is affected by both the electromagnetic interference of the measuring channel and the curvature of the profile of the stationary friction surface of the support roller together with the bending of the frame during the trolley movement.

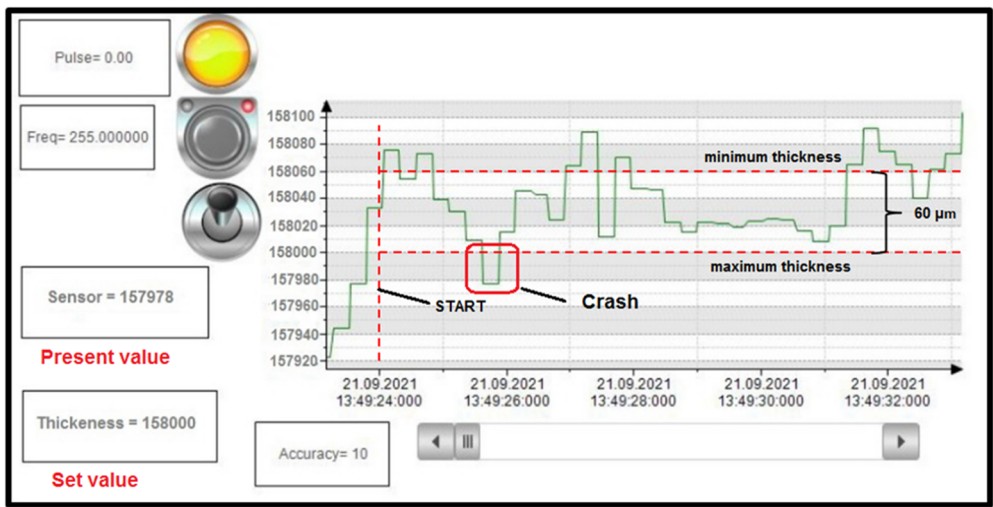

**Figure 4.** Panel for displaying information of the industrial controller.

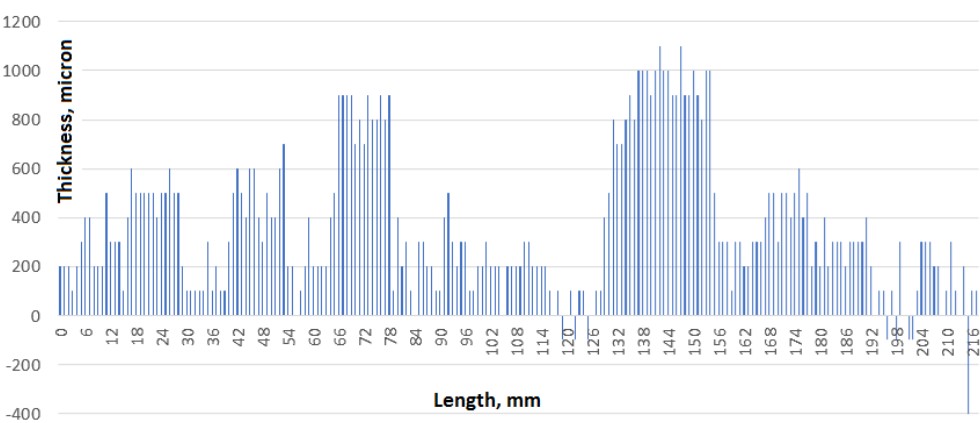

**Figure 5.** Graph of measuring the profile of the material measured thickness.

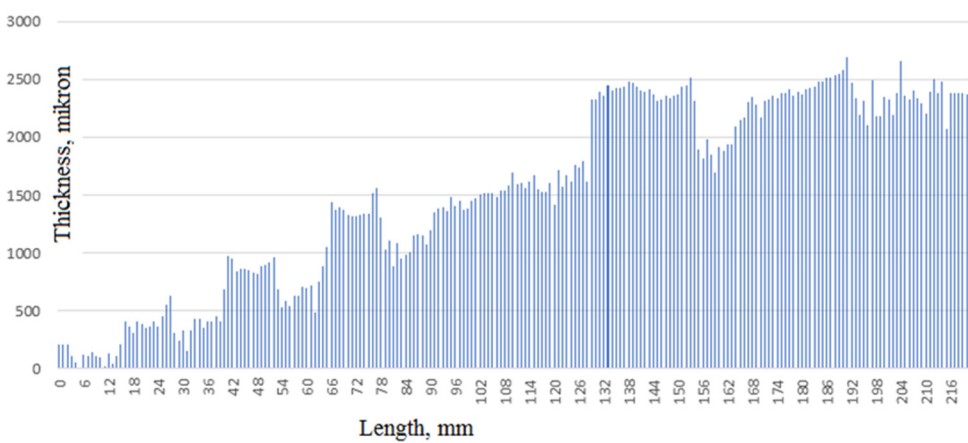

**Figure 6.** Graph of measuring the profile of the material thickness alongside with the geometric curvature of the base surface.

Thus, the interference distorting the measurement result is caused by:

- 50 Hz pickups on the sensor signal wires;

- high-frequency interference caused by the operation of the power supply voltage converters and the operation of the controller modules;
- low-frequency vibrations of the trolley drive structure;
- the static error caused by the curvature of the measured surface.

All the above sources of signal distortion require the use of mathematically processing the measurement results in order to isolate the useful signal from the background noise.

The used method of determining the film thickness as the difference between the distance from the laser sensor to the sliding surface of the film and the distance from the sensor to the film surface gives correct results only at a constant distance to the sliding surface. The geometric curvature of the sliding surface or the bending of the frame profile, when the carriage moves, distorts the result of calculating the thickness.

Figure 7 shows the profile of the base surface on which the measured profile of the material was placed in the experiment. The values of the geometric curvature of the base surface will introduce significant distortions into the profile of the measured material thickness.

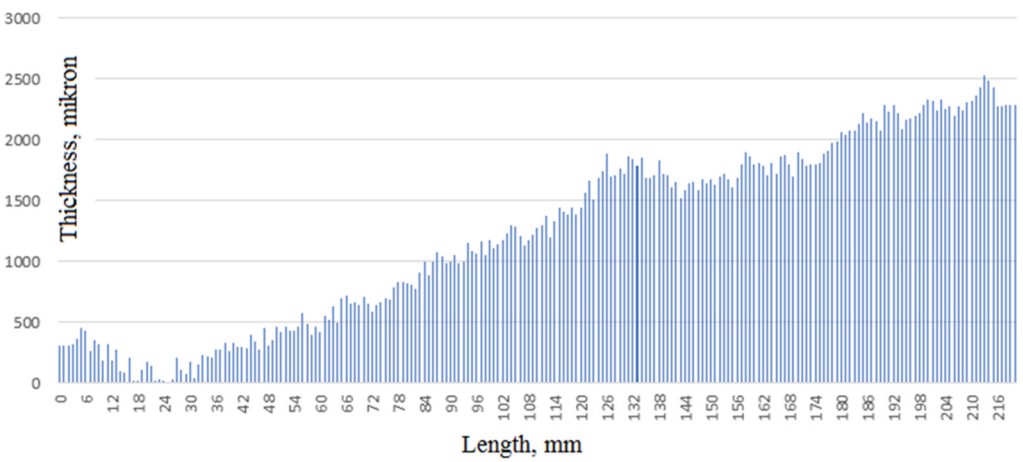

**Figure 7.** Graph of the base surface profile.

The result of the difference between the distances from the sensor to the geomembrane and the sliding surface is shown in Figure 8, where the final profile of thickness measurement is highlighted with the red line.

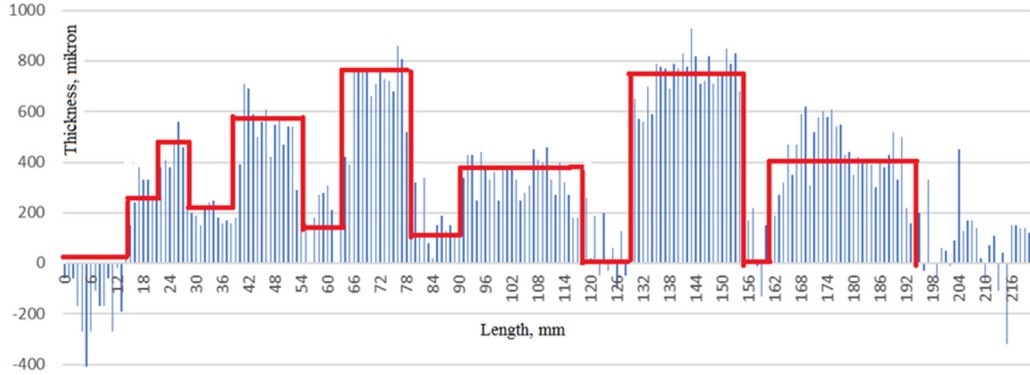

**Figure 8.** Desired graph for measuring the profile of the material thickness.

The technical implementation of such a differential measurement is realized automatically by the simultaneous operation of two distance sensors. In this mode, there is a simultaneous measurement of the distance from the sensor to the geomembrane and from

the sensor to the sliding surface. The desired result is the difference in the readings of the two sensors.

If the initial profile of the sliding surface is recorded in the controller memory, one sensor can be dispensed with by subtracting the distance from the sensor to the geomembrane from the value of the saved surface profile.

Mathematical processing of measurement results includes averaging the measured values, for example, using the "moving average" method that smooths out short-term fluctuations and highlights major trends or cycles. Alternatively, when moving the measuring carriage with the sensor, for example, every 2 mm along the geomembrane, one can measure the distance ten-fold, after which, eliminating gross errors, one can find the arithmetic mean of the measured value.

### 4. Conclusions

In this study, the authors obtained and tested the following results:

1.  The original design solutions were applied in the development of the hardware and software complex mechanisms that allow measuring and controlling the thickness of the polymer geomembrane layers over the entire width of the shaft excluding deformation of the geomembrane due to the ingress of foreign bodies during extrusion (cuts).
2.  A system of monitoring the quality of the geomembrane in real-time has been developed.
3.  The possibilities of analyzing the measured parameters in the database of the developed automated control system have been presented.
4.  Implementation and consideration of the aforementioned features when monitoring the geomembrane thickness with a laser distance sensor makes it possible to make a workable measuring system, while the cost of such a geomembrane thickness monitoring system, according to preliminary calculations, will be USD 10,000, which is 10 times lower than analogs in accordance with the accuracy ensuring.

**Author Contributions:** Conceptualization, supervision, T.N.; methodology, M.B., A.S.; formal analysis, O.Z.; investigation, E.K., L.D.; writing—original draft preparation, E.D., T.N., A.S.; writing—review and editing, E.D., O.Z., M.B., and T.N.; project administration, T.N.; funding acquisition, M.B. and T.N. All authors have read and agreed to the published version of the manuscript.

**Funding:** This research was funded by the Science Committee of the Ministry of Education and Science of the Republic of Kazakhstan according to the contract No.64-3 (15 May 2020) «Design and creation of prototypes of automated production control systems, remote monitoring and diagnostics of the microclimate in polymer agricultural sleeves for storing grain» (Grant No. AP08052553).

**Institutional Review Board Statement:** Not applicable.

**Informed Consent Statement:** Not applicable.

**Data Availability Statement:** This research is published for the first time. Full research data will be published in the Research Report «Design and creation of prototypes of automated production control systems, remote monitoring and diagnostics of the microclimate in polymer agricultural sleeves for storing grain», grant No. AP08052553, registration number 0120PK00094 on the web portal of the National Center for State scientific and technical expertise of the Republic of Kazakhstan (https://is.ncste.kz) in December 2021.

**Conflicts of Interest:** The authors declare no conflict of interest.

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
