# Peer review of "Developing a Measuring System for Monitoring the Thickness of the 6 m Wide HDPE/LDPE Polymer Geomembrane with Its Continuous Flow Using Automation Equipment"

_applsci, doi:10.3390/app112110045_

Round 1
Reviewer 1 Report
Dear authors,
thanks for the article, I consider it to be up to date and of the highest interest. But at the same time there are some points that I need to highlight.
I would like to pay your attention at the References. To my mind the number of references is too small, which means the research of methods and theory was not deep enough, which can lead to the low interest of publication of this paper.
I would also recommend to check the English language and style and shorten the title of the article a bit.
with best regards
Author Response
Response to Reviewer 1 Comments
Dear Reviewer,
We express our deepest gratitude for your remarks and comments!
Point 1: I would like to pay your attention at the References. To my mind the number of references is too small, which means the research of methods and theory was not deep enough, which can lead to the low interest of publication of this paper.
Response 1: The Reference was supplemented by relevant publications that were used in the study but not indicated earlier
Point 2: I would also recommend to check the English language and style and shorten the title of the article a bit.
Response 2: The translation of the article into English was corrected by a professional translator with experience in translating technical literature. The title of the article has also been corrected.
Best regards, authors!
Reviewer 2 Report
This paper proposes a measurement system that uses automated equipment to control the thickness of polymer geomembrane. The innovation lies in the use of developed hardware and software complex mechanisms to control the thickness of the polymer geomembrane along the entire width of the shaft. And presents the possibilities of analyzing the measured parameters in the database of the developed automated control system.
I have suggestions for the following questions:
- HDPE.LDPE only appears in the title and abstract. Can it be deleted? If it is necessary, you can consider replacing ‘.’ With ‘·’or’-‘, otherwise readers may mistake it for two sentences. And the full name of HDPE.LDPE should be given in the article, such as High Density Polyethylene (HDPE).
- The English language and style require extensive editing. ‘The authors ’appeared many times in the article, it is recommended to replace.
3.Line72 .The measuring system has also become widespread. Please add the researcher or institution in the sentence.
- Line186 .Give the basic parameters of the product used .
5.Line209. Delete () and change to a sentence ,keep the same format as below.
6.Line225. It is recommended to replace ‘Let's consider...’ with other sentences.
7.Line280 .Figure 6, there is a shadow on the left edge. Some other pictures in the article have borders and some do not.
- Results. Please analyze the data of the chart, and give the specific value of the error.
- References. Most of the reference documents are website materials, it is recommended to add documents to maintain the authoritativeness of the article.
As the conclusion, due to these problems in this paper, a major revision is suggested.
Author Response
Response to Reviewer 2 Comments
Dear Reviewer,
We express our deepest gratitude for your remarks and comments!
Point 1: HDPE.LDPE only appears in the title and abstract. Can it be deleted? If it is necessary, you can consider replacing ‘.’ With ‘·’or’-‘, otherwise readers may mistake it for two sentences. And the full name of HDPE.LDPE should be given in the article, such as High Density Polyethylene (HDPE).
Response 1: The measuring system was developed and tested for extruders that produce exactly the type of geomembranes indicated in the title of the article and it’s HDPE/LDPE, therefore it is necessary to emphasize this in the title of the article. In the text of the article an indication of the type of HDPE/LDPE geomembranes has been added. The name of the geomembrane types has been changed to HDPE/LDPE. Explanation of abbreviations HDPE/LDPE is given in the Abstract.
Point 2: The English language and style require extensive editing. ‘The authors ’appeared many times in the article, it is recommended to replace.
Response 2: The translation of the article into English was corrected by a professional translator with experience in translating technical literature.
Point 3: Line72 .The measuring system has also become widespread. Please add the researcher or institution in the sentence.
Response 3: Researcher’s names have been added.
Point 4: Line186 .Give the basic parameters of the product used .
Response 4: The basic parameters of Frame 1 have been added.
Point 5: Line209. Delete () and change to a sentence ,keep the same format as below.
Response 5: The parentheses () have been removed, the proposal has been reduced to the recommended format.
Point 6: Line225. It is recommended to replace ‘Let's consider...’ with other sentences.
Response 6: Sentence has been corrected.
Point 7: Line280 .Figure 6, there is a shadow on the left edge. Some other pictures in the article have borders and some do not.
Response 7: Figure 6 corrected. Figure 4 shows a screenshot of the display of the measurement system software, so it has boundaries.
Point 8: Results. Please analyze the data of the chart, and give the specific value of the error.
Response 8: specific values of the thickness deviation are added in Figure 4 and in the description to the figure.
Point 9: References. Most of the reference documents are website materials, it is recommended to add documents to maintain the authoritativeness of the article.
Response 9: The Reference was supplemented by relevant publications that were used in the study but not indicated earlier.
Best regards, authors!

Round 2
Reviewer 1 Report
Dear Authors,
I suggest accepting this article after your revision.
with best regards,
Yury Klochkov
Reviewer 2 Report
This paper proposes a measurement system that uses automated equipment to control the thickness of polymer geomembrane. The innovation lies in the use of developed hardware and software complex mechanisms to control the thickness of the polymer geomembrane along the entire width of the shaft. And presents the possibilities of analyzing the measured parameters in the database of the developed automated control system.